# Reducing Hallucinations in Large Vision-Language Models via Latent Space Steering

**Sheng Liu**   **Haotian Ye**   **James Zou**
Stanford University
{shengl,haotianye,jamesz}@stanford.edu

## Abstract

Hallucination poses a challenge to the deployment of large vision-language models (LVLMs) in applications. Unlike in large language models (LLMs), hallucination in LVLMs often arises from misalignments between visual inputs and textual outputs. This paper investigates the underlying mechanisms of hallucination, focusing on the unique structure of LVLMs that distinguishes them from LLMs. We identify that hallucinations often arise from the sensitivity of text decoders to vision inputs, a natural phenomenon when image encoders and text decoders are pre-trained separately. Inspired by this, we introduce Visual and Textual Intervention (VTI), a novel technique designed to reduce hallucinations by steering latent space representations during inference to enhance the stability of vision features. As a task-agnostic test-time intervention, VTI can be easily applied to any problem without additional training costs. Extensive experiments demonstrate that it can effectively reduce hallucinations and outperform baseline methods across multiple metrics, highlighting the critical role of vision feature stability in LVLMs.[1]

## 1 Introduction

Large Vision-Language Models (LVLMs) (Liu et al., 2023b; Zhu et al., 2023; Ye et al., 2023; Li et al., 2023a; Dai et al., 2023; Gong et al., 2023; Bai et al., 2023b) have demonstrated impressive performance across various tasks such as image captioning (Li et al., 2023b; Wang et al., 2023b), visual question answering (Lee et al., 2024; Wang et al., 2024a), medical treatment planning (Liu et al., 2024b), and many more. LVLMs take advantage of both powerful vision encoders, such as CLIP (Radford et al., 2021) as well as text decoders, which are often pretrained large language models. LVLMs can present sophisticated understandings of vision information by mapping them to the language domain that LLMs can process.

Despite their remarkable success, LVLMs still encounter numerous challenges that impede their applications in real-world tasks, among which hallucination (Liu et al., 2023a; Lovenia et al., 2023; Leng et al., 2023b; Liu et al., 2024a; Deng et al., 2024; Zhu et al., 2024) is one prominent concern (Gunjal et al., 2023; Li et al., 2023c). Hallucination in LVLMs emerges when the generated textual responses include inaccurate descriptions of the input image (Li et al., 2023c), such as mentioning non-existing objects or characters, or providing incorrect spatial relations. Such kind of cross-modality inconsistency arises from a different mechanism from that in standard LLMs, where vision information is missing and hallucination comes solely from the linguistics level. In LVLMs, the sequential relation of information flow from the vision encoder to the text decoder can distinctly contribute to hallucination, a potentially essential question to be answered.

While a few existing studies have explored the underlying mechanism in this regard, leading to different conclusions such as statistical pre-training bias (Agarwal et al., 2020; Agrawal et al., 2016; Goyal et al., 2017), over-reliance on language prior (Leng et al., 2023b; Lee et al., 2023; Zhibo et al., 2023; Han et al., 2022; Wu et al., 2022), and biased feature learning (Zhu et al., 2024; Huang et al., 2023; Yue et al., 2024), they focus mainly on issues that do not distinguish between LVLMs and LLMs. That is, the mechanisms of hallucination they present do not take into consideration the particular structure of LVLMs that differs from LLMs.

---

[1]Code is available at https://github.com/shengliu66/VTI.

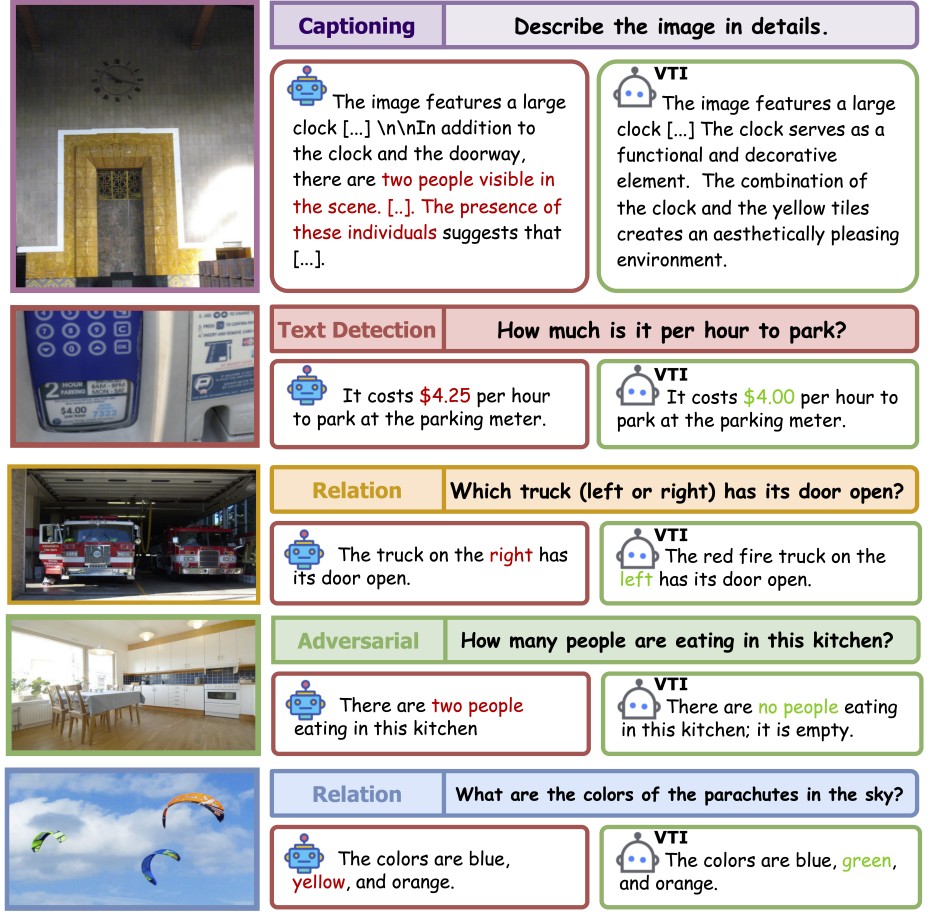

Figure 1: Illustration of the effects of VTIon mitigating hallucination with LLaVA-1.5 as the backbone. Hallucinated contents generated by the original model are marked in red. In contrast, VTI results in less hallucination across different categories of questions. Examples are obtained from MMHAL-Bench (Sun et al., 2023) and CHAIR (Rohrbach et al., 2018)

This paper explores the mechanism of hallucination based on the sequential relation from vision encoder to the text decoder: we demonstrate that hallucinations can stem from the sensitivity to the unstability of the vision encoders. Similar to conventional neural networks in computer vision, an ideal vision encoder should output stable features when the semantics of images remain unchanged, e.g., after we perturb them mildly. This is especially the case for LVLMs where the image encoder and the text decoder are pre-trained separately and minimally fine-tuned together. The sensitivity of the text decoder to image features can exacerbate the misalignment between both modalities and results in hallucinations. We comprehensively present and analyze the correlation between vision feature stability and model hallucinations in Section 3.

A naive way to reduce model hallucination is to smooth vision features by averaging across multiple perturbations of the original image every time we have a new query. Unfortunately, this not only requires multiple forward pass of the model which is extremely inefficient but also introduce extra noises to the original features, leading to worse preservation of information. To better leverage the correlation and reduce hallucinations without introducing additional side effects, inspired by works on representation engineering for LLMs (Liu et al.; Cao et al., 2024; Luo et al., 2024; Zou et al., 2023), where LLM's behavior is altered by editing the latent features, we pre-compute the direction that feature averaging has brought in the latent space and apply them to edit the latent features of

any new query. The pre-computed directions in the latent space capture the underlying changes as we perturb the query image and average vision embeddings across perturbations. This simple yet effective test-time intervention on vision, when combined with intevention on text, allows our proposed method, namely **Visual and Textual Intervention** (VTI), to mitigate vision-language hallucinations. Importantly, we use a fixed 50 examples to pre-compute all intervention directions, and applied consistently across all benchmarks, suggesting VTI is task and dataset agnostic.

In summary, we (1) investigate the mechanism of hallucination by focusing on the sequential relationship between the vision encoder and text decoder, uncovering a correlation between the stability of visual features and hallucinations of LVLMs, (2) propose a novel method, VTI, which guides LVLMs toward less hallucinated outputs by steering them in the latent space, and (3) conduct extensive experiments to benchmark VTI against baseline methods. The leading results across multiple metrics demonstrate the high effectiveness of VTI in reducing hallucinations. We hope that our work can shed light on the critical role of feature stability in LVLMs and inspire future research to develop more effective strategies for reducing hallucinations.

## 2 RELATED WORK

**Large Vision-Language Models.**    The success of Large Language Models (LLMs)(Gilardi et al., 2023; Touvron et al., 2023; Tay et al., 2022; Raffel et al., 2020; Brown et al., 2020; Chowdhery et al., 2022; Taori et al., 2023; Chiang et al., 2023; Bai et al., 2023a) has fueled significant advancements in Large Vision-Language Models (LVLMs)(Liu et al., 2023b; Dai et al., 2023; Zhu et al., 2023; Li et al., 2023a; Ye et al., 2023; Bai et al., 2023b). By incorporating visual encoders and feature projectors (Li et al., 2019; Sun et al., 2019; Wang et al., 2022; Li et al., 2022), LVLMs have achieved notable improvements across a variety of multimodal tasks, including image captioning (Li et al., 2023b), visual question answering (Zhang et al., 2023a), and image segmentation (Lai et al., 2024). However, similar to LLMs, LVLMs are susceptible to generating hallucinations (Li et al., 2023c), which hinder their robustness and reliability in real-world applications. Our work seeks to address this by mitigating hallucinations in LVLMs, thereby enhancing their utility and dependability in practical settings.

**Hallucination in Vision-Language Models and LVLMs.**    In natural language processing, hallucination refers to the generation of false or nonsensical content (Ji et al., 2023; Zhang et al., 2023b; Shi et al., 2023). This issue has recently drawn attention in multimodal models, where hallucinations can significantly impair performance (Wang et al., 2023a; Zhao et al., 2023; Huang et al., 2023; Yue et al., 2024). To counter this, various methods have been proposed, primarily involving additional training to align models with ground truth (Gunjal et al., 2023; Liu et al., 2023a; Sun et al., 2023; Yin et al., 2023; Zhou et al., 2023; Zhai et al., 2023; Yue et al., 2024). However, such approaches often suffer from practical limitations, such as the need for extensive training and additional data. In response, training-free methods have gained traction. These approaches rely on techniques like self-feedback correction (Lee et al., 2023; Yin et al., 2023), leveraging auxiliary models for knowledge integration (Wan et al., 2024; Deng et al., 2024; Zhao et al., 2024; Yang et al., 2024; Kim et al., 2024), and refining the model's decoding process (Huang et al., 2023; Leng et al., 2023a; Favero et al., 2024; Zhang et al., 2024; Wang et al., 2024b). These methods typically focus on adjusting the text decoder but are limited in addressing hallucinations originating from visual components.

In contrast, our approach targets hallucination reduction by directly steering the latent feature space, impacting both the vision encoder and text decoder. This strategy enables us to mitigate hallucinations arising from both vision and text-centric sources, making our method more comprehensive than those focused solely on refining the text output.

## 3 MECHANISM OF HALLUCINATION IN LVLMS

A large vision language model typically consists of a vision encoder and a large language model as a text decoder. Given an image $v$, a text input $x$ such as a question or prompt about the image, the vision encoder first extracts visual information from the image as vision feature vectors $V = \{v_1, v_2, \ldots, v_n\}$. These vision features are then projected into the same embedding space as the text input, and both the vision and text embeddings are concatenated and passed to the text decoder

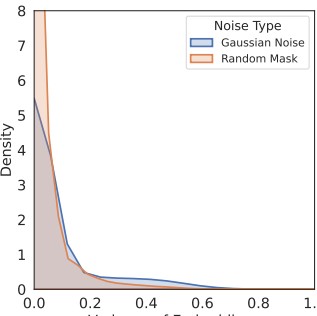 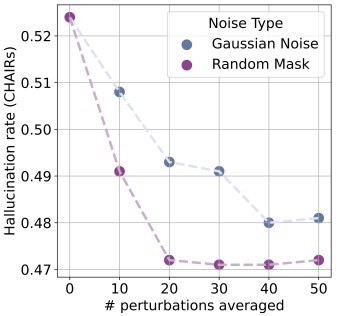 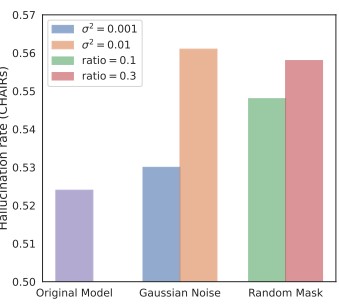

Figure 2: (Left) Distribution of vision feature stability when different types of noise are injected into the raw images. The $x$-axis represents the variance of features across 50 perturbations, and the $y$-axis represents the frequency. (Middle) Illustration of the correlation between object hallucination and vision feature stability. Averaging vision features across multiple perturbed images reduces hallucination (measured by CHAIR, details in Section A) as the number of perturbations averaged increases. (Right) Noise alone tends to increase hallucination, suggesting that the reduction of hallucination is not due to the noise itself but to the averaging process across perturbations.

to generate text outputs auto-regressively. While LVLMs are known to hallucinate, the underlying causes of this phenomenon remain unclear, particularly in relation to the vision encoder's role in the process. In this section, we explore how the stability of vision features impacts hallucination.

We begin by highlighting a fundamental relationship between the vision and text components of large vision-language models: the output of the vision encoder serves as the input to the language decoder. This sequential connection implies that the stability of the vision features plays a crucial role in the model's outputs and can influence the occurrence of hallucinations. To examine the connection between feature stability and object hallucination, we perturb raw images with various types of noise and analyze the variance in the resulting feature distributions. Ideally, noise that does not alter an image's semantic content should have minimal impact on the vision features or the model's output, provided the vision encoder is robust and trained to capture semantic information effectively.

However, as shown in the left figure of Figure 2, while most vision features remain stable, approximately 15% exhibit significant variance, resulting in a long-tailed distribution. These unstable features are closely tied to hallucinations, as the model becomes overly sensitive to these features, leading to inaccuracies in its outputs. This connection is further demonstrated in the second figure of Figure 2: averaging vision features across multiple noise-injected images reduces hallucination as the number of perturbations increases, regardless of the type of noise injected. Importantly, as shown in the right figure, this reduction is not due to the noise itself but to the averaging process across perturbations, while noise alone tends to increase hallucination.

Although feature averaging can mitigate hallucination, it introduces significant drawbacks. Since vision encoders are trained on clean data, perturbing images compromises information extraction, leading to a loss of detail. Moreover, averaging vision features requires multiple forward passes through the model, greatly increasing computational costs. Thus, our goal is to develop a method that enhances vision feature stability and reduces hallucination efficiently, without sacrificing information or incurring excessive computational overhead.

## 4 METHOD

The experiments in the previous section demonstrate that features averaged across mildly corrupted images are robust and effective in reducing hallucinations, despite the side effects. To avoid this, inspired by works on representation engineering for LLMs (Liu et al.; Cao et al., 2024; Luo et al., 2024; Zou et al., 2023), where LLM's behavior is altered by editing the features in the latent space during inference, we develop a computationally efficient algorithm called **visual and textual intervention** (VTI) that can improve vision feature stability as well as text to image dependancy to reduce hallucination of LVLMs. In particular, we pre-compute the "direction" of more stable features

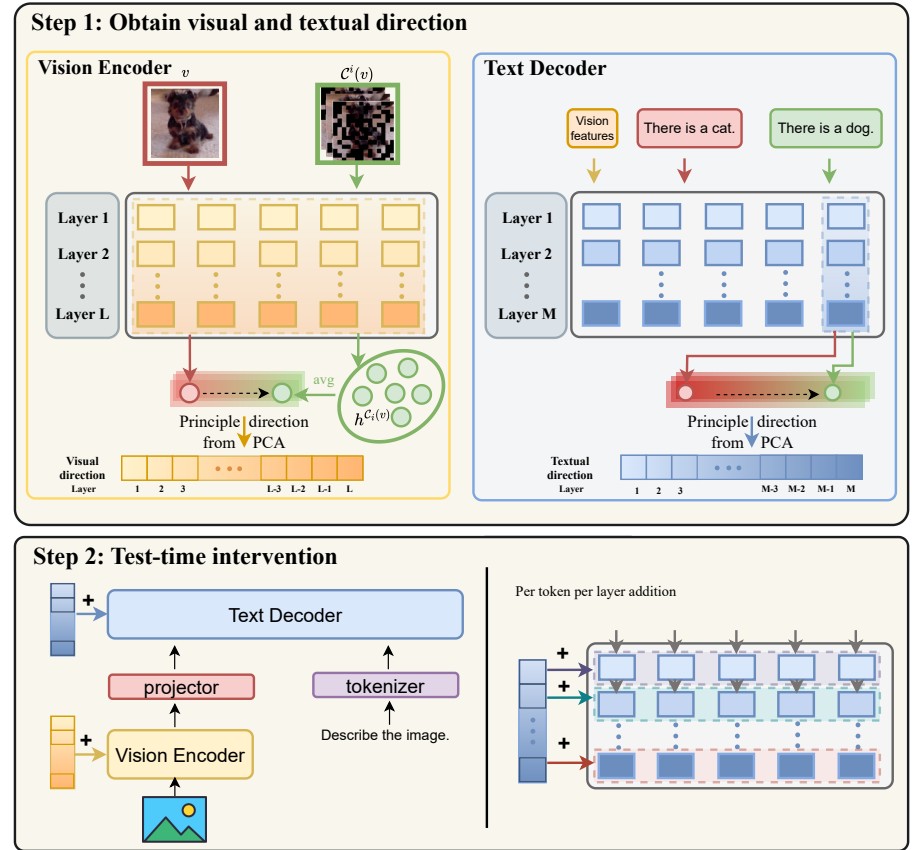

Figure 3: Overview of the proposed algorithm visual and textual test-time intervention (VTI). Given a example set $\{(v_i, x_i, \tilde{x}_i)\}_{i=1}^N$ where $v_i$ is the vision input and $(x_i, \tilde{x}_i)$ is paired captions with and without hallucination, VTI first runs the model on each query $(v_i, x_i, \tilde{x}_i)$ and records all hidden states. It then computes the shifting vectors $d_{l,t}^{\text{vision}}$ and $d_{l,t}^{\text{text}}$ for all layer $l$ and token $t$ according to Section 4. During inference, the vectors are subsequently added to every layer of the vision encoder and text decoder, respectively, when processing a new query. Notice that the vectors are task- and dataset-agnostic, i.e., they are pre-computed using a few samples from one specific task and dataset, and fixed unchanged throughout the entire experiments in our paper.

and then apply them consistently to all query examples during inference to reduce hallucination, without introducing additional training or inference cost. As sometimes hallucination rise from the text decoder, i.e. the LLM, we further obtain a textual direction and apply it to the text decoder to maximize the performance. The overview of the proposed method is illustrated in Figure 3.

Given a vision input $v$, let the latent states of the vision encoder for $v$ be represented as $h_{l,t}^v$, where $l \in \{1, 2, \ldots, L\}$ denotes the layer index in the encoder, and $t \in \{1, 2, \ldots, T\}$ represents the index of the vision tokens.

To enhance the robustness of the latent representation, we leverage random masking to create multiple perturbed versions of $v$. Specifically, we apply $m$ different random masks, $\mathcal{C}_i, \ i = 1, \ldots, m$, to $v$, producing $m$ corrupted versions, $\mathcal{C}_i(v)$. For each perturbed input $\mathcal{C}_i(v)$, the vision encoder generates the corresponding latent states $h_{l,t}^{\mathcal{C}_i(v)}$.

Intuitively, a robust latent embedding for $v$ can be approximated by averaging the embeddings obtained from the perturbed inputs. This averaged embedding is computed as:

$$\bar{h}_{l,t}^v = \frac{1}{m} \sum_{i=1}^m h_{l,t}^{\mathcal{C}_i(v)}.$$

The **visual shifting vector** is then defined as the difference between the robust averaged embedding and the original embedding:

$$\Delta_{l,t}^v = \bar{h}_{l,t}^v - h_{l,t}^v. \tag{1}$$

To make this shifting vector applicable to new image queries, we aim to remove image-specific information from $\Delta_{l,t}^v$, retaining only the general effect introduced by feature averaging. To achieve this, we compute the shifting vectors $\Delta_{l,t}^{v_i}$ for a set of $N$ example images, $\{v_1, v_2, \ldots, v_N\}$. By stacking these vectors into a matrix,

$$[\Delta_{l,t}^{v_1}, \Delta_{l,t}^{v_2}, \ldots, \Delta_{l,t}^{v_N}],$$

we extract the first principal direction of this matrix, denoted as $d_{l,t}^{\text{vision}}$. This principal direction captures the dominant pattern of change introduced by feature averaging (an ablation study of the PCA dimensions is provided in Table 5).

It is worth noting that for each image, the random mask applied to generate the perturbed inputs $\mathcal{C}_i(v)$ is independently sampled, ensuring diversity in the perturbations and improving the generality of the computed shifting vector.

Apart from the vision token shifting, we further introduce the textual shifting vector that steers the latent states of the text decoder when generating model outputs. Obtaining the textual shifting vector is as simple as what previous work proposed in aligning the style of LLMs; Following (Zhou et al., 2023), we curated $N$ image captions without hallucination, denoted as $x$ and adopt GPT model generate the hallucinated version $\tilde{x}$. As a result, we obtain paired captions with and without hallucination. We simply use the original corresponding images $v$ as the visual inputs. We then compute the textual direction for each of the sample as

$$\Delta_{l,t}^{x_i, v_i} = h_{l,t}^{x_i, v_i} - h_{l,t}^{\tilde{x}_i, v_i}, \tag{2}$$

where $h_{l,t}$ stands for the hidden states of the $t$-th text token in layer $l$ when generating the last token of outputs. In particular, since the text decoder is causally modeled, we only use latent states of the last token, i.e. $t =$ the last token. Similarly, we perform PCA to remove extra noise from a specific example choice to obtain the overall direction $d_{l,t}^{\text{text}}$.

We separately apply the visual and textual direction to Intervene vision encoder and the text decoer, i.e. the LLM. Since the vision encoder is not causally modeled, we shift the latent states of all layers of the vision encoder at all token positions in the forward pass:

$$h_{l,t}^v := h_{l,t}^v + \alpha \cdot d_{l,t}^{\text{vision}}, \tag{3}$$

and for the text, we shift the latent states of the text decoder with the textual direction:

$$h_{l,t}^{x,v} := h_{l,t}^{x,v} + \beta \cdot d_{l,t=\text{last token}}^{\text{text}}, \tag{4}$$

As we will demonstrate in the experimental section below, such a combination of vision and language latent space intervention can address various styles of hallucination.

## 5 EXPERIMENTS

In this section, we empirically investigate the effectiveness of VTI in reducing hallucinations. Remarkably, we use 80 examples with paired images from COCO (Lin et al., 2014)), hallucinated and non-hallucinated responses to pre-compute the visual and textual direction and apply them to all tasks, datasets, and queries. This ensures the universality and generalizability of our results. We aim to answer the following questions: (1) Can visual intervention effectively reduce hallucination in LVLMs? (2) Can textual intervention effectively reduce hallucinations in LVLMs? (3) What is the benefit of combining them?

| Model | LLaVA-1.5 | | InstructBLIP | | Qwen-VL | |
|---|---|---|---|---|---|---|
| Method | Accuracy ↑ | F1 Score ↑ | Accuracy ↑ | F1 Score ↑ | Accuracy ↑ | F1 Score ↑ |
| Vanilla | 79.8 | 79.4 | 76.3 | 78.0 | 83.5 | 81.2 |
| VCD | 82.3 | 83.4 | 80.1 | 81.0 | 84.5 | 83.3 |
| OPERA | 84.2 | 83.7 | 79.6 | 80.9 | 84.3 | 82.6 |
| **VTI** | **86.5** | **85.9** | **81.8** | **83.2** | **85.2** | **84.1** |

Table 1: POPE hallucination evaluation results on three different LVLMs. We report the average F1-score and accuracy averaged across three sub-tasks. The complete table can be found in Appendix B.

## 5.1 EXPERIMENTAL SETTINGS

**Datasets.** We evaluate our model on both discriminative and generative datasets, as listed below. More details about the datasets are provided in the appendix. **(a) POPE**: The Polling-based Object Probing Evaluation (Li et al., 2023c) contains 27,000 Yes/No questions about object existence in MSCOCO (Lin et al., 2014), where the task is to judge whether the given object is in the given image (examples are provided in Figure 7). Following existing works, we compute accuracy, precision, recall, and F1 score for each method. **(b) CHAIR**: Caption Hallucination Assessment with Image Relevance (Rohrbach et al., 2018) quantifies object hallucinations in image captions by comparing generated objects to ground-truth objects. Following previous works (Huang et al., 2023; Yue et al., 2024), we randomly select 500 images from the MSCOCO dataset (Lin et al., 2014) and use $CHAIR_I$, $CHAIR_S$, and Recall as evaluation metrics. **(c) MMHAL-Bench** (Sun et al., 2023): This benchmark evaluates LVLMs beyond object hallucination and contains eight question types: object attributes, adversarial objects, comparisons, counting, spatial relations, environment, holistic description, and others. We evaluate the hallucination rate and response informativeness using GPT-4.

**Implementation Details.** We evaluate the effectiveness of our model on three mainstream large vision-language models, including InstructBLIP (Dai et al., 2023), LLaVA 1.5 (Liu et al., 2023b) and Qwen-VL (Bai et al., 2023b) with beam search as the default decoding strategy (denoted as Regular). We also compare our model with state-of-the-art baseline methods: OPERA (Huang et al., 2023) and VCD (Leng et al., 2023b). These methods focus on mitigating object hallucinations by improving the decoding strategy. We perform a grid search for the strength of vision and text vectors where $\alpha, \beta \in \{0.1, 0.2, 0.3, 0.4, 0.5, 0.6, 0.7, 0.8, 0.9, 1.0\}$. For baseline methods, we followed the settings in their papers and released code to ensure a fair comparison. More details are provided in the Appendix A.

## 5.2 EXPERIMENTAL RESULTS

**Results on POPE.** We begin with the most extensively used benchmark for object hallucination. In Table 1, we compare various decoding-based hallucination mitigation methods on the POPE benchmark (Li et al., 2023c), where "Vanilla" stands for the original model. Obviously, VTI outperforms the regular decoding strategies across all LVLMs consistently, resulting in the best accuracy and F1 score. In addition, VTI surpasses state-of-the-art contrastive decoding methods, demonstrating its effectiveness in mitigating object hallucinations. Notice that we average three different sub-tasks in POPE, and the comprehensive results can be found in Appendix B.

**Results on Open-ended Generation with CHAIR Evaluation.** Beyond the "Yes-or-No" discriminative evaluations on POPE, we validate our model on open-ended caption generation using the CHAIR benchmark (Rohrbach et al., 2018). The results in Table 2 demonstrate consistent improvement over the compared methods. Remarkably, visual shifting and textual shifting demonstrate effectiveness on different assessment dimensions: visual shifting is more effective in reducing $CHAIR_I$ that calculates on image-level, while textual shifting is more effective in reducing $CHAIR_s$ that calculates on sentence-level. Combining them (VTI, the last row) helps incorporate their benefits and effectively reduces object hallucinations in generated captions, as evidenced by lower $CHAIR_S$ and $CHAIR_I$ scores. In addition, VTI also maintains the comprehensiveness of the generated captions, as indicated by higher recall scores.

| Model | Method | CHAIR$_S$ ↓ | CHAIR$_I$ ↓ | Recall ↑ | Avg. Len |
|---|---|---|---|---|---|
| | Vanilla | 51.0 | 15.2 | 75.2 | 102.2 |
| | DOLA | 57.0 | 15.9 | 78.2 | 97.5 |
| | VCD | 51.0 | 14.9 | 77.2 | 101.9 |
| LLaVA1.5 | OPERA | 47.0 | 14.6 | 78.5 | 95.3 |
| | **Vision only** | 43.2 | 12.7 | **78.6** | 93.4 |
| | **Text only** | 41.0 | 12.9 | 78.3 | 92.2 |
| | **VTI** | **35.8** | **11.1** | 76.8 | 93.8 |
| | Vanilla | 54.0 | 18.1 | 71.1 | 115.3 |
| | DOLA | 60.0 | 20.1 | 71.5 | 110.8 |
| | VCD | 57.0 | 17.0 | 72.1 | 112.1 |
| InstructBLIP | OPERA | 54.0 | 12.8 | 69.8 | 93.6 |
| | **Vision only** | 49.1 | 12.1 | **72.5** | 104.2 |
| | **Text only** | 48.7 | 14.2 | 72.1 | 98.7 |
| | **VTI** | **43.4** | **11.8** | 70.1 | 105.8 |

Table 2: CHAIR hallucination evaluation results with max new tokens set to 512. Smaller CHAIR corresponds to less hallucination, and higher recall is better. We bold the best results and underline the second-best results. VTIand its variations consistently have the lowest hallucination rate with similar recall.

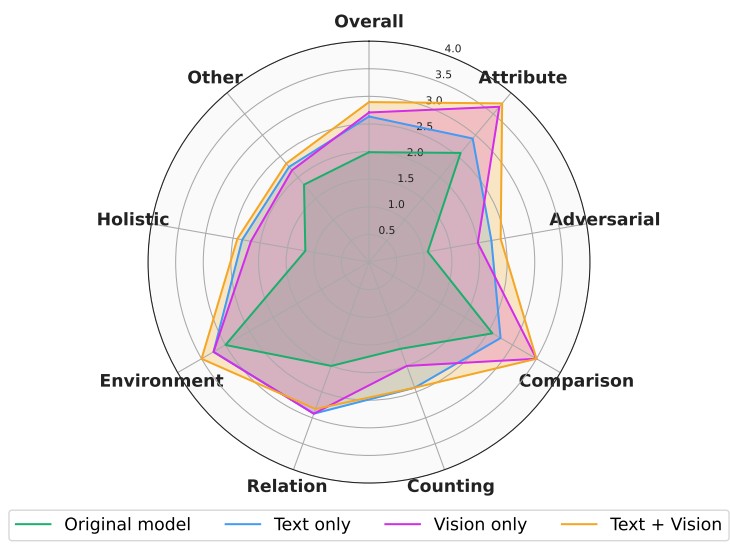

Figure 4: Detailed performance of different models on the eight categories in MMHAL-BENCH (Sun et al., 2023), where "Overall" indicates the averaged performance across all categories. A higher score indicates that the generated response contains fewer hallucinations and more information.

**Results on MMHAL-Bench.** To extensively test our method on various vision tasks, we benchmark VTI on the MMHAL dataset with the evaluations of eight different hallucination types. Comparison with existing methods are shown in Table 3, and a fine-grained ablation is presented in Figure 4, VTI achieves a much higher score (corresponds to less hallucination) across all categories. This underscores its effectiveness in addressing a broader range of multimodal hallucination challenges beyond objects. We also observe that textual and visual shifting are complementary. Visual shifting performs better on vision-centric tasks such as recognizing and comparing attributes of objects. In contrast, textual shifting is good at reducing hallucination of textual-centric tasks such as answering adversarial questions and counting, where relatively more complex language reasoning is required.

## 5.3 ANALYSIS

**Can visual shifting improve feature stability?** As we discussed in Section 3, simply averaging vision embeddings across multiple perturbed images can reduce hallucination. VTI serves as a "soft"

| Method | Average score ↑ | Hallucination rate ↓ | Attribute | Adversarial | Comparison | Counting | Relation | Environment | Holistic | Other |
|---|---|---|---|---|---|---|---|---|---|---|
| Vanilla | 1.99 | 0.62 | 2.58 | 1.08 | 2.58 | 1.67 | 2.00 | 3.00 | 1.17 | 1.83 |
| VCD | 2.69 | 0.58 | 3.25 | 2.17 | 3.00 | **2.42** | 2.58 | 3.25 | **2.42** | **2.42** |
| OPERA | 2.64 | 0.59 | 2.92 | 2.25 | 2.75 | **2.42** | **2.92** | 3.25 | 2.33 | 2.25 |
| VTI | **2.90** | **0.51** | **3.75** | **2.42** | **3.50** | **2.42** | 2.83 | **3.5** | **2.42** | 2.33 |

Table 3: Results on MMHAL-Bench (Sun et al., 2023) with LLaVA-1.5. Our proposed method significantly reduces hallucinations in different categories.

alternative to the naive averaging, and it remains unclear whether the algorithm indeed improve feature stability. To show this, we apply various types of pertubations to images, including random mask, Gaussian blur, random noise, random brightness adjustment, and random elastic transform. For each of the settings, we compute the feature variance (across 100 perturbations) and average across different features as the metric of feature stability. We then compare the feature stability with/without applying VTI in Figure 5. As expected, the vision features obtained from the model with vision intervention exhibit lower variance and better stability, suggesting the effectiveness of the vision direction in smoothing out the vision features.

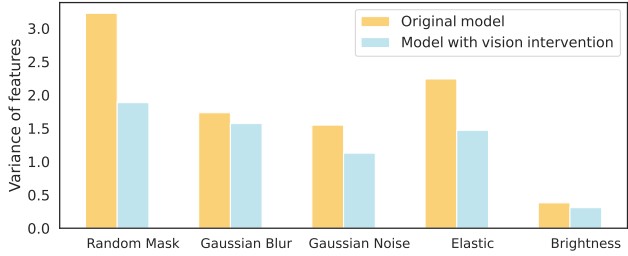

Figure 5: Feature stability is improved across different types of image corruptions with vision intervention.

**Textual intervention increases attention dependency toward images.** It has been observed that LVLMs can occasionally disregard input image's information but treat the next-token prediction as a purely text-based continuation task (Zhu et al., 2024). Text intervention, by pushing the latent space features to the non-hallucination direction, implicitly reduces such text-biased hallucinations. Figure 6 illustrates that textual intervention is able to reduce average attention from generated text to text while increasing attention to vision tokens, suggesting higher reliance on the image when generating text.

**Combining visual and textual intervention can enhance the level of detail in generations.** A straightforward, though unremarkable, approach to reducing hallucinations is to generate less content. However, as illustrated in Figure 6, combining visual and textual interventions offers the advantage of achieving similar hallucination reduction without the need to shorten the generation length.

**Why is visual intervention more effective than simple vision feature averaging?** In Section 3, we illustrate the effectiveness of averaging features to mitigate hallucination, however, adding random noise to the input images and averaging the corresponding features may result in a loss of information. As we demonstrated in Figure 8, performing linear probing on the simple averaged features results in only around 62% classification accuracy while applying visual intervention with different strength $\alpha$ resulting in improvement of visual feature stability (expressed in lower feature variance) without significantly hurting the probing accuracy, suggesting better information preservation.

**Ablation of $\alpha$ and $\beta$.** Eventually, we study the influence of vector strength applied on hidden states. Figure 8 presents the results of an ablation study on $\alpha$ and $\beta$, which controls the strength of visual and textual intervention, respectively. Figure 8 illustrates that $\alpha = \beta = 0$, implying no

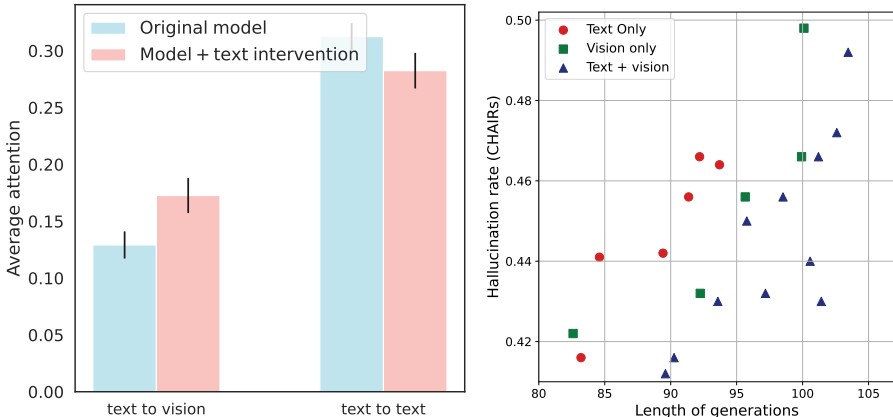

Figure 6: (Left) Textual intervention reduces the self-attention from text to text tokens and increases the self-attention to the vision tokens. (Right) Combining vision and text intervention can achieve similar hallucination rates but with longer generations.

intervention, results in suboptimal performance on CHAIRs. As $\alpha$ and $\beta$ increase, improvements in the hallucination rate are observed, highlighting the effectiveness of the visual and textual intervention.

## 6 CONCLUSION

In conclusion, our exploration into mitigating object hallucinations in LVLMs through latent space steering has yielded promising results. By implementing visual and textual intervention, we have significantly reduced hallucinations without compromising the models' ability to generate detailed and contextually accurate outputs. Our findings underline the importance of robust feature representation and its pivotal role in enhancing model reliability. As LVLMs continue to evolve, we believe the strategies outlined in this paper will serve as foundational steps toward creating more accurate, trustworthy, and efficient systems, fostering broader applicability in real-world scenarios. We are optimistic that future advancements will build on these insights, bridging the gap between human-like understanding and machine-generated interpretations.

### ETHIC AND REPRODUCIBILITY STATEMENT

We propose the visual and textual intervention method to address hallucination issues in LVLMs, thereby enhancing their safety and reliability within the community. Additionally, the datasets utilized for inferring and evaluating our method are publicly accessible, promoting transparency and reproducibility in our research. Furthermore, we will make our code available to the public once accepted, ensuring it is convenient for researchers and practitioners to access and implement.

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

## A  DETAILED EXPERIMENTAL SETTINGS

In all experimental setups, the mask ratio to compute visual direction is set to $0.99$, and we average across 50 random masks. For experiments on CHAIR, to maintain similar lengths of generations, we set $\alpha = 0.4$ for visual intervention only, and similarly, $\beta = 0.4$ for textual intervention only. For VTI. $\alpha = \beta = 0.4$. For experiments on MMHAL-Bench, we adopted $\alpha = 0.9$ and $\beta = 0.9$.

**POPE**[2] We utilize the official benchmark from Li et al. (2023c), which includes 3,000 question-answer pairs for each of the random, popular, and adversarial settings. We use the query

---

[2]https://github.com/RUCAIBox/POPE

template 'Is there a [object] in the image?'. Here, [object] is selected randomly, from the most frequent objects in the dataset, or from objects that frequently co-occur with [object], corresponding to the random, popular, and adversarial settings respectively. We evaluate the performance based on whether the model-generated output contained the ground truth ('Yes' or 'No') using accuracy, precision, recall, and average F1-score.

**CHAIR**[3] We select 500 random images from the COCO Lin et al. (2014) validation set and generate the output using the query "Please Describe this image in detail.". Due to the computational complexity, we restrict the *max new tokens* to 64. Following the M3ID Favero et al. (2024), we report two assessment metrics, $\text{CHAIR}_s$ and $\text{CHAIR}_i$, which calculate the hallucination ratio per sentence and instance as follows:

$$\text{CHAIR}_s = \frac{|\{\text{sentences with hallucinated objects}\}|}{|\{\text{all sentences}\}|}, \text{CHAIR}_i = \frac{|\{\text{hallucinated objects}\}|}{|\{\text{all objects mentioned}\}|}. \quad (5)$$

**MMHAL-Bench**[4] In MMHAL-Bench dataset, 96 image-question pairs, ranging in 8 question categories × 12 object topics are included. The question categories are

- **Object attribute:** LVLMs incorrectly describe the visual attributes of invidual objects, such as color and shape.

- **Adversarial object:** LVLMs answers questions involving something that does not exist in the image, instead of pointing out that the referred object cannot be found.

- **Comparison:** LVLMs incorrectly compare the attributes of multiple objects.

- **Counting:** LVLMs fail to count the number of the named objects.

- **Spatial relation:** LVLMs fail to understand the spatial relations between multiple objects in the response.

- **Environment:** LVLMs make wrong inference about the environment of the given image.

- **Holistic description:** LVLMs make false claims about contents in the given image when giving a comprehensive and detailed description of the whole image.

- **Others:** LVLMs fail to recognize the text or icons, or incorrectly reason based on the observed visual information

Following Sun et al. (2023), we use GPT-4 to evaluate different methods on MMHAL-BENCH and to analyze and rate responses. Prompts we used for MMHAL-BENCH evaluation can be found in (Sun et al., 2023). The rating scales are

- 6, very informative with good analysis or reasoning, no hallucination

- 5, very informative, no hallucination

- 4, somewhat informative, no hallucination

- 3, not informative, no hallucination

- 2, very informative, with hallucination

- 1, somewhat informative, with hallucination

- 0, not informative, with hallucination

and if the rating $< 3$, the generation is considered with hallucination.

---

[3]https://github.com/LisaAnne/Hallucination
[4]https://huggingface.co/datasets/Shengcao1006/MMHal-Bench

# B  ADDITIONAL EXPERIMENT RESULTS

## B.1  DETAILED RESULTS ON POPE

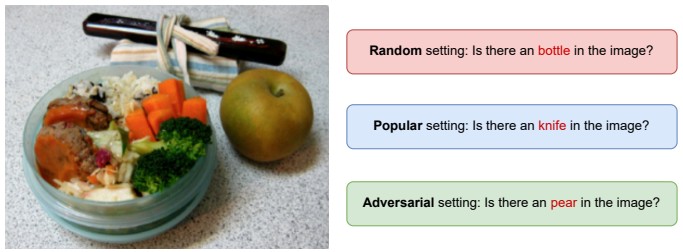

Figure 7: Example questions in different settings of the POPE dataset

We present the detailed performance on POPE in Table 4. VTI exhibits significant performance improvement with LLaVA-1.5, InstructBLIP, and Qwen-VL. These results underscore the effectiveness of VTI in mitigating object hallucination.

## B.2  DETAILED RESULTS ON MMHAL-BENCH

We present the detailed performance on MMHAL-Bench in Table 3presents the detailed performance metrics on MMHAL-Bench. VTI demonstrates substantial improvements across various tasks, consistently outperforming both VCD and OPERA in all question categories except for the 'Other' category. These results highlight the effectiveness of VTI in reducing hallucinations beyond object-specific errors, reinforcing its potential to enhance LVLMs' accuracy in interpreting and analyzing visual content comprehensively.

## B.3  MORE RESULTS ON ANALYSIS

In this section, we provide additional figures (Figure 8) on the analysis section in the main text. Visual shifting better preserves vision information (measured by linear probe accuracy on ImageNet) than naive feature averaging while reducing feature variance.

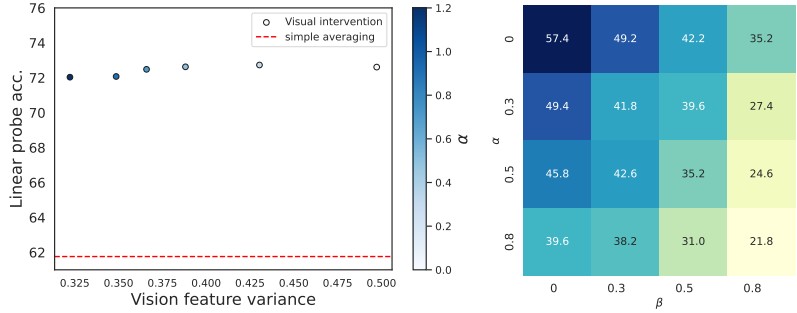

Figure 8: (Left) Trade-off between robustness and information preservation. Visual shifting improves feature robustness while preserving information. $\alpha$ controls the strength of the shifting. (Right) Ablation study on $\alpha$ and $\beta$.

# C  EXAMPLES FOR COMPUTING THE TEXTUAL DIRECTIONS.

To generate hallucinated captions for our experiments, we curated a training set of 50 examples using the methodology outlined in (Zhou et al., 2023). Specifically, we leveraged GPT-3.5's in-context learning capabilities to automatically generate hallucinatory data for refinement. First, GPT-3.5

| Setting | Model | Decoding | Accuracy | Precision | Recall | F1 Score |
|---|---|---|---|---|---|---|
| *Random* | LLaVA1.5 | Regular | 83.49 | 88.84 | 76.76 | 82.36 |
| | | OPERA | 87.53 | **94.52** | 79.80 | 86.53 |
| | | VCD | 86.84 | 87.15 | **86.68** | 86.91 |
| | | **VTI** | **89.50** | 94.38 | 84.00 | **88.89** |
| | InstructBLIP | Regular | 80.42 | 78.93 | 83.21 | 81.01 |
| | | OPERA | 85.07 | 88.39 | 80.73 | 84.39 |
| | | VCD | 84.11 | 84.20 | 85.36 | 84.78 |
| | | **VTI** | **86.37** | **88.83** | **85.68** | **87.22** |
| | Qwen-VL | Regular | 84.33 | 95.98 | 71.67 | 82.06 |
| | | OPERA | 85.13 | **95.54** | 74.13 | 83.48 |
| | | VCD | 85.53 | 95.17 | 74.87 | 83.81 |
| | | **VTI** | **86.73** | 93.67 | **78.80** | **85.59** |
| *Popular* | LLaVA1.5 | Regular | 79.98 | 82.47 | 76.76 | 79.51 |
| | | OPERA | 84.21 | 88.00 | 79.80 | 83.70 |
| | | VCD | 82.65 | 87.15 | 80.59 | 83.74 |
| | | **VTI** | **87.36** | **91.61** | **82.27** | **86.69** |
| | InstructBLIP | Regular | 76.09 | 73.22 | 82.94 | 77.78 |
| | | OPERA | 78.33 | 73.85 | 87.73 | 80.19 |
| | | VCD | 79.94 | 77.84 | 83.33 | 80.49 |
| | | **VTI** | **81.86** | **80.17** | **85.68** | **82.83** |
| | Qwen-VL | Regular | 84.17 | 96.13 | 71.20 | 81.81 |
| | | OPERA | 84.73 | **96.52** | 73.13 | 83.21 |
| | | VCD | 85.63 | 94.28 | 75.87 | 84.08 |
| | | **VTI** | **85.67** | 92.06 | **78.06** | **84.48** |
| *Adversarial* | LLaVA1.5 | Regular | 76.03 | 76.11 | 76.80 | 76.45 |
| | | OPERA | 80.88 | 82.16 | 79.76 | 80.94 |
| | | VCD | 77.31 | 73.43 | **86.47** | 79.42 |
| | | **VTI** | **82.57** | **84.33** | 80.01 | **82.11** |
| | InstructBLIP | Regular | 72.37 | 68.78 | 83.06 | 75.24 |
| | | OPERA | 75.50 | 70.49 | **87.73** | 78.17 |
| | | VCD | 76.32 | 73.24 | 84.08 | 78.29 |
| | | **VTI** | **77.29** | **74.09** | 85.67 | **79.46** |
| | Qwen-VL | Regular | 81.93 | 91.30 | 70.60 | 79.62 |
| | | OPERA | 82.93 | **91.15** | 73.31 | 81.26 |
| | | VCD | 82.19 | 90.14 | 75.53 | **82.19** |
| | | **VTI** | **83.13** | 88.20 | **77.67** | 82.16 |

Table 4: Experimental results on POPE (Li et al., 2023c).

| Method | Hallucination Rate (%) |
|---|---|
| LLaVA-1.5 (base) | 61.46 |
| VTI (PCA dim = 1) | 51.04 |
| VTI (PCA dim = 3) | 51.04 |
| VTI (simple average) | 57.96 |

Table 5: VTI performance with different PCA dimensions. Experiments are conducted on MMHAL-Bench.

was prompted to produce a list of objects that frequently co-occur with the ones mentioned in a given description. Then, using LLaVA-1.5, we generated descriptions for the 50 training images, incorporating a randomly selected word from the "co-occurring objects" list and another from the "uncertain objects" list. This process allowed us to create a rich dataset of hallucinated captions. For further details on the prompts, refer to (Zhou et al., 2023). Figure 9 shows an example of a generated caption.

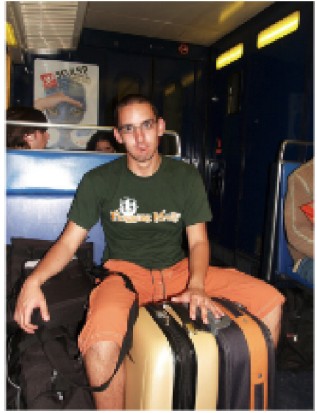

**Original Caption:** The image shows a young man sitting on a pile of luggage while traveling on public transportation, likely a passenger train or a bus. He has a puzzled look on his face as he tries to manage his belongings on this crowded journey. There are multiple suitcases, a handbag, and a backpack nearby, indicating that the man has a considerable amount of luggage with him. Apart from the man sitting on his luggage, there are a few other people in the scene as well, some sitting on benches while others stand in the space. The two benches available are located on either side of the man sitting with his luggage. Additionally, there are handbags placed on the floor in the same area, suggesting that other passengers also have their belongings with them.

**Generated Hallucinated Caption:** The image shows a young man with a puzzled look on his face as he tries to manage his multiple suitcases, handbag, and backpack while sitting on a pile of luggage during a crowded journey on public transportation. Other passengers are seen with their own handbags and belongings nearby. One of the passengers is seen holding a water bottle, while it is uncertain what the man in front of the young man is doing or why he is smiling.

Figure 9: Illustration of the generated hallucinated captions for textual direction computing.

