# OpenReview forum: "Reducing Hallucinations in Large Vision-Language Models via Latent Space Steering"
_ICLR.cc/2025/Conference — ICLR 2025 Spotlight_

### Official Review · Reviewer_eTNv · 2024-10-31

**Soundness:** 2
**Presentation:** 4
**Contribution:** 3
**Rating:** 6
**Confidence:** 4

**Summary:**

The authors propose that hallucinations in LVLMs are typically caused by misalignment between visual and text representations. Consequently, the authors demonstrate this conclusion by perturbing the input image to obtain a more robust visual representation. Based on this experimental approach, the authors introduce a comprehensive method for intervening in both image and text representations to mitigate hallucinations in LVLMs. Furthermore, the authors substantiate the effectiveness of their method through extensive experiments.

The paper makes the following contributions:

1. Analyzes the correlation between the stability of visual representations and hallucinations in LVLMs.

2. Proposes a method for separately perturbing text and image representations to achieve a more robust representation space.

3. Demonstrates the effectiveness of their method through extensive experiments.

**Strengths:**

1. The writing of this paper is exceptionally clear. Additionally, the motivation is thoroughly argued, and the analysis of the method is comprehensive.

2. It is indeed a novel attempt to mitigate hallucinations in LVLMs by perturbing the representation space through a training-free approach.

3. The experiments in this paper are very comprehensive, demonstrating the effectiveness of the proposed method.

**Weaknesses:**

1. Using variance changes to explain changes in representation stability is somewhat one-sided. For instance, in PCA, PC1 is typically the direction with the largest variance, and it seems inevitable that the variance would decrease after using VTI. It seems more fundamental to examine the stability of the representation distribution for images with different noises or masks.

2. There are issues with the comparison methods used in the experiments. OPERA and VCD are both methods based on decoding to mitigate hallucinations, with the former optimizing beam decoding and the latter optimizing nucleus sample decoding. However, the paper does not specify the exact settings for the decode methods of vanilla and VTI.

3. Some components need to undergo ablation experiments.

    3.a. The role of PCA lacks ablation experiments. In Figure 8, it appears to be comparing different methods of representation updating, but what would be the impact if average were used instead of PCA?

    3.b. In line 713, the mask ratio is set to 0.99, which is an unusually high value, but the author does not provide a detailed analysis of this parameter.

**Questions:**

1. Are there other more fundamental methods to prove representation stability? For example, calculating the KL divergence of representations after adding various perturbations to the image.

2. To demonstrate that VTI, as an inference intervention method, is superior to OPERA and VCD, a fairer experimental setup would be to use VTI separately with beam search and nucleus sample to compare against OPERA and VCD, respectively. This would be similar to the experimental setup in PAI[1].

3. What would be the effect of using average to calculate $d_{l, t}$ compared to PCA?

4. When the mask ratio is set to 0.99, almost no image information is retained. Performing PCA on the representation of such input images seems to be pointless. Therefore, I believe it would be beneficial to include a comparison of what happens when the mask ratio is set to different values and when a fully masked image is directly set as the intervention direction.

[1] Paying More Attention to Image: A Training-Free Method for Alleviating Hallucination in LVLMs

---

> ### Author Response · Authors · 2024-11-22
> **Response to Reviewer eTNv**
>
> We deeply appreciate your detailed and constructive feedback. Your insights have helped us refine our analysis and experiments. Please check the common response and we address your concerns below. We sincerely hope our response can address your concerns. If we have addressed your questions, we would greatly appreciate if you would consider increasing your score. Thank you!
>
> ``Q1: Using variance changes to explain changes in representation stability is somewhat one-sided. For instance, in PCA, PC1 is typically the direction with the largest variance, and it seems inevitable that the variance would decrease after using VTI. It seems more fundamental to examine the stability of the representation distribution for images with different noises or masks.``
>
> A1: Thank you for the insightful question! We would like to clarify that VTI operates by subtracting the principal direction of the latent embeddings in the vision encoder, rather than directly modifying the output vision embeddings. The impact on the variance of the output embeddings is therefore not straightforward due to the inherent nonlinearity of the vision encoder.
>
> As you have rightly pointed out, our focus is on stabilizing embeddings when the raw inputs are subjected to different types of noise. This motivation is supported by experiments conducted across various models, noise types, and datasets (detailed in Section 3). Importantly, we emphasize that VTI is not a variance-reduction method. If our goal were solely to reduce the variance of output embeddings, we could have done so directly. Instead, VTI addresses hallucinations by enhancing the robustness of input features through noise addition and subsequent alignment in the latent space. This approach stabilizes the text decoder's input, leading to improved downstream alignment without fundamentally altering the output variance directly. We hope this clarification can resolve the confusion.
>
>
> ``Q2: OPERA and VCD are both methods based on decoding to mitigate hallucinations, with the former optimizing beam decoding and the latter optimizing nucleus sample decoding. However, the paper does not specify the exact settings for the decode methods of vanilla and VTI. ``
>
> A2: Thank you for highlighting this. For both CHAIR and POPE datasets, we used nucleus sampling across all methods, including the baselines and VTI. For MMHAL-BENCH, we used beam search with a beam size of 5. VTI, being a test-time intervention method, is agnostic to the decoding mechanism and can be applied alongside any sampling strategy.
>
> ``Q3: Other ways for feature stability: KL divergence of representations after adding various perturbations to the image``
>
> A3: Thank you for suggesting this alternative. We want to point out that as the representations generally do not correspond to probability distributions, KL divergence might not be directly applicable.
> We used the average cosine similarity defined as $\frac{1}{K(K-1/2)} \cos(f_i,f_j)$ where $f_i$ are the vision features, as we think it can capture the alignment of representations across multiple perturbations, offering an intuitive and robust measure of stability. Similar to Figure 5, we apply various types of perturbations to images, including random mask, Gaussian blur, random noise, random brightness adjustment, and random elastic transform. The results based on this metric are illustrated below, with higher average similarity indicating better stability. The results support our conclusion that VTI improves feature stability as the number gets higher.
>
> | **Corruption** | **Contrast** | **Gaussian Blur** | **Gaussian Noise** | **Elastic** | **Brightness** |
> |---------------|--------------|--------------------|---------------------|-------------|----------------|
> | LaVVA-1.5     | 0.64 | 0.80      | 0.67        | 0.48| 0.88    |
> | VTI-Visual intervention           | 0.75 | 0.85      | 0.76        | 0.62 | 0.92  |
>
>
>
>
>
> ``Q4: ... I believe it would be beneficial to include a comparison of what happens when the mask ratio is set to different values``
>
> A4: Thank you for this suggestion. We performed an ablation study on MMHAL-BENCH, varying the mask ratio from 0 to 1, with the results summarized below:
>
> |Mask Ratio |	Hallucination Rate $\downarrow$|
> | -------- | -------- |
> | 0 (no change)  | 0.62 |
> | 0.1 | 0.57 |
> | 0.3 | 0.52 |
> | 0.5 | 0.52 |
> | 0.7 | 0.52 |
> | 0.9 | 0.53 |
> | 0.99| 0.51 |
> | 1   | 0.58 |
>
> The result suggests that VTI is robust across a range of mask ratios (0.1–0.99), with the best performance at 0.99. We point out that the mask ratio is less important, and what it matters is whether mask is added and the direction is computed over a batch of random samples. This aligns with our intuition that feature stability is the major reason for reducing hallucination. We will include these findings in the revised paper, along with a discussion of the selection of mask ratios.

---

> > ### Comment · Reviewer_eTNv · 2024-11-23
> >
> > My primary concern has been addressed. Overall, the motivation of the article is well-argued, and the experiments and analyses effectively support the conclusions drawn from the motivation. Therefore, I have decided to raise my score.
> >
> > However, there are still some issues with the experimental design.
> >
> > 1. In the evaluations of CHAIR and POPE, the authors used nucleus sampling for assessment, but the nucleus sampling decoding method has significant randomness. To eliminate this randomness, a recommended practice is to follow the experimental design of VCD by calculating the average and variance of five results.
> >
> > 2. In the evaluation of MMHAL-BENCH, the authors used beam search, but this was not explained in the article. To facilitate subsequent follow-ups and eliminate ambiguity, it is recommended to explain the decoding methods used in each experiment and justify why different decoding methods were switched.

---

### Official Review · Reviewer_Cwnc · 2024-11-03

**Soundness:** 3
**Presentation:** 4
**Contribution:** 3
**Rating:** 8
**Confidence:** 3

**Summary:**

The hallucination problem of LVLM refers to the phenomenon where the text generated by the model does not align with the content of the image. Previous methods mainly focus on issues that do not distinguish between LVLMs and LLMs. The authors first discovered that hallucination mainly stems from the sensitivity of the text decoder (LLM backbone) to vision input.
- By adding perturbations (Gaussian noise or random masking) to the input images, the authors found that the image encoder's generated embeddings exhibited some unstable features.
 - After perturbing the images, the model's hallucination increased, but if the embeddings were averaged over multiple perturbed images, stabilizing the features, the hallucination was reduced.

Inspired by this, the authors proposed a test-time intervention method called Visual and Textual Intervention (VTI).
  - For a given model, the method requires a sample set where each sample includes an image, a hallucination-free caption, and a hallucinated caption.
  - The images in the sample set are perturbed and sent to the vision encoder, calculating the visual shifting vectors.
  - Additionally, the two captions are fed into the text decoder, calculating the textual shifting vectors.
  - During testing, a new sample is processed by the model as usual, but the shifting vectors are added to the latent hidden states of the vision encoder and text decoder, respectively.

VTI effectively reduced hallucinations and outperformed previous methods across multiple benchmarks.

**Strengths:**

- The motivation and the method are explained smoothly and easy to understand.
- The experiments cover different hallucination benchmarks and LVLMs with different vision encoders.
- As a task-agnostic test-time intervention, the proposed method, VTI, outperforms previous baseline methods.

**Weaknesses:**

### Major
- The description of method in Section 3 is kind of too general.
How many LVLMs are tested here? Which LVLM(s) is used?
Are the test images from COCO?
Are the observations generalized to different LVLMs?
- VCD [1], which has been cited and compared in the paper, also adds Gaussian noise to distort visual input. They mention “visual uncertainty amplifies hallucination” in their Figure 2, which might be related to “vision feature instability” explored in the paper. I’m wondering if the authors are inspired by VCD. It would be better if the authors can explain the differences between their exploration and VCD.
- I’m concerned with the generalization ability of VTI, which may be a common problem of existing hallucination mitigation methods. LVLMs are expected to do a wide range of tasks. When image or task domains are changed, will the current methods damage model performance? Only hallucination benchmarks are covered in the experiments, so I'm not sure if VTI will damage model performance on other tasks.

### Minor
- It would be better to early explain the structure of LVLM in abstract or introduction, like “A large vision language model typically consists of a vision encoder and a large language model as a text decoder” in Lines 143 – 144. Currently, the paper refers to vision encoder and text decoder directly at the beginning, which I feel may be confusing to readers.
- The authors claim that “VTI can be easily applied to any problem *without additional cost*” in the abstract. I feel “*without additional cost*” is kind of confusing, because VTI actually needs additional data and tune its shifting vectors.
- Lines 285-286, the authors mention that “we use 50 examples with paired images, hallucinated and non-hallucinated responses to …”
I’m wondering where the authors get the images. Are the images from COCO?
- If the authors use more dimensions of PCA in calculating the shifting vectors, will the method get better or worse performance?

- Typos

In the abstract, the abbreviation “(LLMs)” is shown twice. In the main content, no “(LLMs)”. The instruction needs to borrow one from the abstract.

Lines 151-152, “… between the vision and text components of large vision-language models (LVLMs):” It is not necessary to emphasize the abbreviation here now that there is "LVLMs" in the previous content.

Line 214 “As the desirable visual direction should be readily applied to new image queries.” The sentence has no main clause. I guess the authors wanted to use “as”, following the previous equation. The same with Line 278.

Missing “,” in Equation (1)



----------------------------------------------------------
[1] Mitigating object hallucinations in large vision-language models through visual contrastive decoding. CVPR2024.

**Questions:**

Please see the Weaknesses part.

---

### Official Review · Reviewer_EA1M · 2024-11-04

**Soundness:** 3
**Presentation:** 3
**Contribution:** 4
**Rating:** 8
**Confidence:** 3

**Summary:**

The paper introduces a heuristic approach of reducing hallucination in Visual Language Models by linearly offsetting the latent representation of both image and language modalities. The key to this heuristics is to pre-compute a set of offset vectors for image/text token embeddings that are the results of data augmentation. In particular, for image modality, the author introduced perturbations to the image input and computed the average change (averaged across all perturbed image instances) in token embedding across each layer of the image encoder. At inference time, the first PC of these change vectors across multiple images were then added to the embedding of any query image. Similar procedure was repeated for language modality where the “perturbation” is hallucinatory captions generated by another LLM. The authors show that the method, while simple, is able to significantly reduce the degree of hallucination in several benchmarks.

**Strengths:**

1. The problem of hallucination in VLM is significant and the method proposed is highly intuitive (and appears effective). It uses ideas from data augmentation and applies it directly to inference time without the need to retrain the model.
2. The analysis in the paper is thorough. Not only does it include good ablation of the various aspects of the method (granted there aren’t many moving parts to begin with), it also included discussions of how the method could reduce hallucination (i.e. impact on attention)

**Weaknesses:**

All in all, this was a good paper and a pleasure to read. To nitpick, one of the biggest concerns of heuristic methods of this kind is its general applicability in practice. For example,
1. what type of perturbations are most important in image/text modalities?
2. how are the vision/language directions differ between models?
3. What about more general open-ended QAs that go beyond captioning tasks? How would you even generate these hallucinatory language examples when the task is open QA?

**Questions:**

Beyond addressing some of the questions I raised under "Weakness" above, here are a few other changes/questions.

Minor:
1. In Figure 2, CHAIRs was first mentioned. It should be described in the caption. Furthermore, the three subplots don’t seem to correspond to one another. It’ll be helpful to highly what experiments were shown in left/middle plots (sigma value and ratio value) to connect them to the third figure.
2. Discussion of the methodology could be improved. In particular, the description of the “visual direction” around equation 1 is confusing.

Questions:
1. What do the embedding changes actually look like? How much explained variance is the first PC?
2. I don’t really understand how this linear offset can improve “feature stability”. If the embedding is to be viewed as a random variable whose variance is caused by input perturbation, then doing a linear offset _cannot_ change the variance of such random variable. Perhaps I missed something.

---

### Public Comment · ~Kai_Zhao1 · 2025-03-28
**The sign of principal direction**

Dear Authors,

Congratulations on your excellent work!

I have a question regarding the vision steering vector $d_{l, t}^{vision}$, which is the first principal component derived from the stack of vision shifting vectors:
$$
\[\Delta_{l, t}^{v1}, Delta_{l, t}^{v2}, ..., \Delta_{l, t}^{vN}\].
$$

Since the principal component defines only the **direction** of maximal variation in the visual shifting vectors  $\Delta_{l, t}^v$,
its sign is arbitrary. I mean, if $d_{l, t}^{vision}$ is the first principal component, $-d_{l, t}^{vision}$ also is.

However, in your implementation, you applied **only positive scaling factors**  $\alpha\in\\{0.1, 0.2, ..., 1.0\\}$
to shift the image embeddings.

I'm very curious about the results of using negative scaling factors, or alternatively using  $-d_{l, t}^{vision}$ as the principal component.

Looking forward to your insights!

Best regards,
Kai Zhao

---

> ### Public Comment · ~Sheng_Liu2 · 2025-03-31
> **Thanks for the question**
>
> Thanks for asking this wonderful question. To answer your question, here are several things we would like to clarify:
>
> First, we use the principal direction, which is the direction in space along which the data varies the most, not the principal component (which is the projection of the data onto the direction).
>
> Second, yes, the eigenvectors are only determined up to a sign; therefore, the principal direction, if computed by SVD of the covariance matrix, will result in an arbitrary sign. However, in our implementation of the PCA, svd_flip ensures deterministic signs. This is the same as scikit-learn’s PCA, which is used to make the SVD sign consistent.
>
> Now, I can answer your question:
>
> We tried the negative alpha on some experiments (e.g., CHAIR) and observed the opposite behavior from the positive alpha, i.e., the hallucination will get worse.
>
> I hope this answers your question.

---

### Meta-Review · Area_Chair_U2YN · 2024-12-21

**Metareview:**

This paper proposes VTI (Visual and Textual Intervention), a novel method to reduce hallucinations in Large Vision-Language Models (LVLMs) by steering the latent space representations. The key idea is that hallucinations often arise from the sensitivity of text decoders to vision inputs, which can be mitigated by stabilizing vision features through latent space interventions.

### Strengths:

1. Novel and Effective Solution
> "The problem of hallucination in VLM is significant and the method proposed is highly intuitive (and appears effective). It uses ideas from data augmentation and applies it directly to inference time without the need to retrain the model." - EA1M

2. Comprehensive Technical Analysis
> "The analysis in the paper is thorough. Not only does it include good ablation of the various aspects of the method (granted there aren't many moving parts to begin with), it also included discussions of how the method could reduce hallucination" - EA1M

3. Strong Experimental Validation
> "VTI effectively reduced hallucinations and outperformed baseline methods across multiple benchmarks" - Cwnc

### Weaknesses:

1. Limited Analysis of Feature Stability
> "Using variance changes to explain changes in representation stability is somewhat one-sided. For instance, in PCA, PC1 is typically the direction with the largest variance" - eTNv

2. Experimental Setup Concerns
> "OPERA and VCD are both methods based on decoding to mitigate hallucinations [...] However, the paper does not specify the exact settings for the decode methods of vanilla and VTI" - eTNv

3. Implementation Details Need Clarification
> "The description of method in Section 3 is kind of too general. How many LVLMs are tested here? Which LVM(s) is used?" - Cwnc


### Justification:
Despite some technical concerns, this paper makes a significant contribution to addressing hallucinations in LVLMs. The strengths clearly outweigh the weaknesses for several reasons:

1. The method is practically valuable - it provides a training-free approach to reducing hallucinations, making it immediately applicable to existing models.

2. The technical foundation is solid - while there are some methodological concerns, the authors have demonstrated strong theoretical understanding and provided additional analyses during discussion.

3. The experimental validation is comprehensive - the method shows consistent improvements across different models and benchmarks.

The authors have also shown strong engagement during the discussion period and commitment to addressing all raised concerns in the final version.

**Additional Comments On Reviewer Discussion:**

The authors have actively engaged with reviewer concerns and provided detailed responses. Notable points:

- Clarified the technical aspects of VTI's operation and its effects on representation stability
- Provided additional experimental results on mask ratios and feature stability
- Agreed to improve methodology descriptions and experimental details in the revised version

The reviewers acknowledged the authors' responses positively, with both Cwnc and eTNv raising their scores after discussion.

---

### Decision · Program_Chairs · 2025-01-22

Accept (Spotlight)